# Radiation-Associated Angiosarcoma of the Breast and Chest Wall Treated with Thermography-Controlled, Contactless wIRA-Hyperthermia and Hypofractionated Re-Irradiation

**DOI:** 10.3390/cancers13153911

**Published:** 2021-08-03

**Authors:** Markus Notter, Emanuel Stutz, Andreas R. Thomsen, Peter Vaupel

**Affiliations:** 1Radiation Oncology, Lindenhofspital Bern, 3012 Bern, Switzerland; 2Members of the Swiss Hyperthermia Network, 5000 Aarau, Switzerland; emanuel.stutz@insel.ch; 3Department of Radiation Oncology, Inselspital, Bern University Hospital, University of Bern, 3010 Bern, Switzerland; 4Department of Radiation Oncology, University Medical Center, University of Freiburg, 79106 Freiburg, Germany; andreas.thomsen@uniklinik-freiburg.de (A.R.T.); vaupel@uni-mainz.de (P.V.); 5German Cancer Consortium (DKTK), Partner Site Freiburg and German Cancer Research Center (DKFZ), 69120 Heidelberg, Germany

**Keywords:** radiation-associated angiosarcoma of the breast, secondary angiosarcoma, re-irradiation, wIRA-hyperthermia, superficial hyperthermia

## Abstract

**Simple Summary:**

This retrospective study reports on 10 patients with radiation-associated angiosarcoma of the breast and chest wall treated during the past decade. In this rare disease local control is highly dependent on the extent of surgery. Further treatment options are urgently needed. Re-iradiation in combination with localized hyperthermia should be considered for adjuvant and definitive treatment of nonresectable radiation-associated angiosarcomas. The presented hypofractionated re-irradiation schedule with 5 × 4 Gy once per week immediately following wIRA-hyperthermia is a promising option to further reduce the radiation dose recommended so far. This could reduce side effects without compromising local control.

**Abstract:**

Background: Radiation-associated angiosarcoma of the breast (RAASB) is a rare, challenging disease, with surgery being the accepted basic therapeutic approach. In contrast, the role of adjuvant and systemic therapies is a subject of some controversy. Local recurrence rates reported in the literature are mostly heterogeneous and are highly dependent on the extent of surgery. In cases of locally recurrent or unresectable RAASB, prognosis is very poor. Methods: We retrospectively report on 10 consecutive RAASB patients, most of them presenting with locally recurrent or unresectable RAASB, which were treated with thermography-controlled water-filtered infrared-A (wIRA) superficial hyperthermia (HT) immediately followed by re-irradiation (re-RT). Patients with RAASB were graded based on their tumor extent before onset of radiotherapy (RT). Results: We recorded a local control (LC) rate dependent on tumor extent ranging from a high LC rate of 100% (two of two patients) in the adjuvant setting with an R0 or R2 resection to a limited LC rate of 33% (one of three patients) in patients with inoperable, macroscopic tumor lesions. Conclusion: Combined HT and re-RT should be considered as an option (a) for adjuvant treatment of RAASB, especially in cases with positive resection margins and after surgery of local recurrence (LR), and (b) for definitive treatment of unresectable RAASB.

## 1. Introduction

Radiation-associated (secondary) angiosarcoma of the breast (RAASB) is a highly aggressive tumor that develops in previously irradiated areas. It occurs due to the utilization of adjuvant radiotherapy (RT) mostly applied in the setting of breast-conserving treatment in breast cancer. c-Myc amplification in RAASB is characteristic and differentiates RAASB from primary angiosarcoma of the breast [1,2]. Awareness of RAASB as a late side effect of RT for breast cancer is essential to facilitate an early diagnosis. However, reddish-livid, flat skin alterations may attract the attention of patients or physicians rather late, and diagnosis is unfortunately often delayed. Since RAASB can easily be confounded with a local hematoma, a skin biopsy should be performed for any suspicious lesion arising on a previously irradiated breast [3].

Recently, several population-based studies [4,5,6], systematic literature reviews and meta-analyses [7,8,9], as well as retrospective single-institution studies [2,10,11] have been published. Epidemiological data from these publications are quite consistent and can be summarized as follows: RAASBs develop in 0.1% of all breast cancer patients treated with adjuvant RT in primary breast cancer care, occurring—on average—8 years after RT, at a mean patient age of around 70 years. The incidence of RAASB is increasing, most likely due to routinely performed adjuvant RT after breast-conserving surgery and due to prolonged survival of breast cancer patients [12].

In contrast to data on incidence, prognostic data from recent publications are extremely heterogeneous and differ considerably from data published several years ago. Five-year overall survival (OS) in population-based surveys ranges from 22.5% in the SEER database analysis of the National Cancer Institute (NCI) in the USA [13], 41% in the Netherlands [4], 50.5% in Italy [6], and 69% in Finland [5]. Including single-institution and systematic literature reviews, the local recurrence rate (LRR) ranges from 35% [10], and 59% [7], to 66% [14]. Median recurrence-free survival (RFS) ranges from 4 months [15], 10 months [16], 23 months [4] up to 36 months [10]. Amongst others, this heterogeneity might be due to low patient numbers including a large variation of tumor stages and prognostic factors, missing treatment guidelines, and selection bias of case reports.

So far, there are no uniform treatment recommendations for RAASB. Surgery is accepted as the basic approach [5,17], wherein the extent of surgery is of major importance. Nevertheless, the risk of local recurrence is high [14]. The role of adjuvant therapies and the treatment strategy for recurrent and unresectable RAASBs remain unclear [18,19]. Due to its radiation-induced etiology, there is a general reluctance of applying re-RT. To address this challenge, modified fractionation schedules such as hyperfractionation [20] or hypofractionation are used, with the latter combined with concurrent hyperthermia (HT) as a radiosensitizer [21,22]. Radiosensitizing effects of HT include (a) an improved oxygenation as a consequence of increased blood flow at times of re-RT [23,24], (b) depletion of glutathione levels [25], and (c) inhibition of DNA repair [26]. In addition, there is evidence that mild HT (39 °C–43 °C) can activate anti-tumor immune responses (e.g., [27]). The radiosensitizing effects allow for a significant reduction of the total re-RT dose [28]. Combined thermography-controlled, water-filtered infrared-A (wIRA), contactless, superficial HT with hypofractionated re-RT reveals excellent clinical effects in pre-irradiated locally recurrent breast cancer (LRBC) providing high patient comfort and low toxicity [29]. This hyperthermia technique guarantees large treatment fields, tissue temperatures >40 °C up to 15 mm, and only low toxicity (mostly grade 1) due to permanent temperature measurements all over the body surface avoiding any burns [30]. In the present retrospective analysis, we report the results of a cohort of RAASB patients treated with the same treatment schedule of combined wIRA-hyperthermia and hypofractionated re-irradiation (HT/re-RT). We discuss the results in relation to other previously published treatment schedules and their outcomes.

## 2. Patients and Methods

Nine consecutive patients presenting with RAASB from 08/2011–06/2015 at the Hôpital Cantonal, La Chaux-de-Fonds (Switzerland) and from 06/2015 to 12/2020 at the Lindenhofspital, Bern (Switzerland), and one patient from 01/2019 to 12/2020 at the University Medical Center, University of Freiburg (Germany), have been retrospectively analyzed. Ethics votes were not required for retrospective analyses since patients have been treated in standard routine use. All patients gave informed consent to use their anonymized data for scientific evaluation and publication.

Patients had previously been treated with adjuvant RT after breast conserving surgery and adjuvant hormonal treatment. One patient had additionally received adjuvant chemotherapy. The initial stage of breast cancer had been favorable, with lymph node involvement in only one patient. Nine out of 10 patients had been treated using conventional RT-fractionation with tangential fields comprising photons (6–15 MV linear accelerator), and one patient had been treated with ^60^Co. Nine patients had received a tumor bed boost, with seven of them using 9–12 MeV electrons (Table 1).

Median latency period of RAASB was 6 years (range: 4.5–14) (Table 2). Clinical observations such as skin type, hair color, the presence of benign and RT-related skin alterations were also recorded. Five out of 10 patients presented with ruby points (cherry hemangiomas), and 2 patients with fibrosis (thickening of the skin) (Table 2).

Tumor extensions at time of presentation at the RT department were divided into 4 categories: I = postoperative after R0, II = postoperative after incomplete resection (inappropriate resection margin, R1, or R2 respectively), III = macroscopic lesions, limited to the ipsilateral chest wall, not surpassing midsternal line, midaxillary line, clavicle or lower costal margin, and IV = macroscopic lesion with far-reaching extension, reaching the contralateral chest wall, supraclavicular fossa, abdominal wall or the back. In addition, depth of infiltration as a putative prognostic factor was classified as follows: a = infiltration limited to cutis and subcutis, and b = deep infiltration into the muscle layer (Figure 1). None of the patients presented with distant metastases from RAASB.

### 2.1. Patient Groups

According to their different histories, patients were divided into 4 groups A–D (Table 3):

Group A: 2 patients (No. 1 and 2) presented directly after surgery of the RAASB, No. 1 with R2, No. 2 with R0 resection status.

Group B: 3 patients (No. 3, 4, and 5) presented after surgery of a recurrent RAASB, No. 3 and 4 with R1, No. 5 with R0 resection status.

Group C: 2 patients (No. 6 and 7) presented with aggressive tumors developing a local RAASB recurrence within 3–4 weeks after primary radical surgery despite R0 resection with preoperative biopsy mapping. In both patients the option of another extensive surgery was abandoned due to the almost immediate recurrence with rapid extension over the chest wall.

Group D: 3 patients (No. 8, 9, and 10) presented with large-sized unresectable recurrences of RAASB, patients No. 8 and 9 in a highly palliative situation. Patient No. 8 had been heavily pretreated (e.g., 2 resections, 7 chemotherapies) and presented after the 3rd RAASB recurrence with a tumor infiltrating into the muscle layer, and No. 9 presented with a large tumor extending to the back. Patient No. 10 presented in a good general condition.

### 2.2. Treatment

Treatment planning was based on CT scan, former radiological investigations, e.g., MRI and PET-CT and clinical descriptions to define properly lateral tumor spread and deep infiltration. RT- and HT-volumes were prescribed with margins of 5 to 10 cm around all visible lesions whenever possible. That means that RT-fields, mostly using electrons, were quite large and mainly limited by the machines’ capability. Electrons were chosen because RAASB is considered a skin disease occurring in cutaneous/subcutaneous layers. The treatment protocol consisted of weekly water filtered infrared-A (wIRA) superficial HT for 45–60 min (hydrosun^®^TWH1500, Hydrosun Medizintechnik, Müllheim, Germany, see Figure 2D), immediately followed by re-RT. The contactless energy deposition allows for continuous thermography as well as visual control of the entire treatment field, which is of crucial importance for critical skin conditions, e.g., skin transplants and mesh grafts (Figure 2C,F). In addition, this technique produces very large treatment fields [30], mostly even larger than radiation fields in order to cover completely the defined treatment volume. The computer-based, closed feedback system of this device was set at a maximum skin surface temperature of 43.0 °C (Figure 2E). This results in tissue temperatures >40 °C at a tissue depth up to approximately 15 mm, and >39 °C up to a depth of 25–30 mm [30]. Patients No. 1–9 were treated with 4 Gy once per week for a total dose of 20 Gy, as described in detail later [28,29,30]. Eight patients were irradiated with 6–9 MeV electrons and a preheated bolus. Patient No. 8 had a combined photon–electron plan to cover deep infiltration. Patient No. 10 received a conventionally fractionated re-RT with 50 Gy (25 fractions with 2 Gy/fraction with photons) with an electron boost up to 60 Gy which was combined with 6 weekly hyperthermia sessions, applied immediately before RT.

## 3. Results

### 3.1. Treatment Outcome

Both patients of Group A (adjuvant HT/re-RT after primary surgery of RAASB) did not develop a recurrence during follow-up (FU), with patient No. 1 (Figure 3 A–D) being observed for 67 months before being lost to FU (LFU), and patient No. 2 for 37 months still currently under observation today.

In Group B (adjuvant HT/re-RT after surgery of recurrent RAASB), patient No. 3 with R1 resection developed a re-recurrence at the border after 6 months which was directly retreated with the same HT/re-RT schedule, resulting in complete remission (CR) (Figure 3A–F). Afterwards, the patient received palliative chemotherapy with paclitaxel due to distant metastasis and developed a new local progression in- and outside the former treatment fields. The patient died 20 months later. Patient No. 4, who initially had an R1 resection, developed three consecutive re-recurrences, the first after 3 months. All re-recurrences were retreated with the same HT/re-RT schedule (5 × 4 Gy, which was repeated three times in total), each time resulting in CR. This patient is currently (45 months after surgery of the first recurrence of RAASB) under observation with slow progression. Documentation is shown in Figure 4A–L. Patient No. 5 with R0 resection is still alive after 10 months, and currently has no evidence of disease (NED).

In Group C (macroscopic recurrence 3–4 weeks after primary radical surgery) patient No. 6 achieved a CR and has been locally controlled for 16 months, with NED. Patient No. 7 has just finished the treatment.

In Group D (unresectable RAASB recurrences) patient No. 8 did not respond to HT/re-RT and died after 1 month. Patient No. 9 achieved a PR with good palliative effect and survived for 7 months. The clinical situation is presented in Figure 5A,B. Patient No. 10 is alive with NED for 18 months. In contrast to the other patients of this group, this patient had a tumor extension limited to the ipsilateral chest wall and was in good general condition. This led to the decision to retreat her with HT and conventionally fractionated re-RT.

### 3.2. Toxicity

Acute and chronic toxicity, according to CTCAE v5.0, are presented in Table 4. Acute toxicity occurred in nine out of 10 patients, with patients presenting with erythema grade 1 (*n* = 7), and hyperpigmentation grade 1 (*n* = 6), respectively. No thermal skin damage was observed by any of the 10 patients. Chronic grade 1 toxicity occurred in two patients: Patient No. 4 who received four series of HT/re-RT (with partially overlapping treatment fields) showed hyperpigmentation and extensive new telangiectasia (Figure 4L), and patient No. 10 who developed distinct hyperpigmentation.

## 4. Discussion

Given the low incidence of RAASB, there is no standard of care, especially regarding adjuvant therapies. For the outcome of initial RAASB treatment, quality and extent of surgery seem to be decisive. Management of the disease in specialized centers may significantly improve outcome [32]. Larger surgical margins are associated with improved survival [5,11,33]. Radical surgery, including resection of all pre-irradiated skin areas, followed by flap reconstructions can significantly decrease the risk of LR, and can lead to exceptionally high OS rates [2,10,11]. However, radical surgery is associated with a signi-ficant increase in postoperative complication rates compared to a more conservative skin resection [11]. In individual cases, the mutilation due to the resection of large skin areas followed by flap reconstruction, must be weighed against limited medical operability of older patients as well as the individual demand of the patient [34]. Pre-clinical and first clinical data on the use of targeted therapies and immunotherapy in the treatment of angiosarcoma have extensively been discussed by Cao et al. [35].

Data on the efficacy of adjuvant and neoadjuvant chemotherapy (CT) as well as adjuvant RT in the treatment of first RAASB manifestation are inconsistent [5,36]. Some authors deny the need for adjuvant RT by presenting good local control rates after radical surgery alone [5,10,11], whereas other authors emphasize the high risk of local recurrence [14,20]. The risk of RAASB recurrence might be caused by wide-spread and occult microscopic metastases within the skin, extending from the macroscopic tumor.

There is a comprehensible reluctance of applying RT to a tumor with radiation-induced etiology. In one multivariate analysis of a populations-based study, adjuvant RT seemed to be associated with worse OS [13]. In contrast, no impact on OS, but significantly better RFS, was found in systematic literature reviews [7,9]. Several authors state that their data do not allow for any recommendation pro or contra adjuvant RT (e.g., [6]).

Smith et al. [20] applied hyperfractionated, accelerated radiotherapy (HART) with atypically generous radiation field margins of 5–10 cm, pre- or postoperatively. They reported a median OS of 7 years, and a 5-year OS of 79%. Notably, surgery was included whenever feasible, with the resection of all pre-irradiated tissues [20]. Efficacy of this method was confirmed by observations of Donovan et al. [37].

The two patients in our cohort who received adjuvant re-RT/HT after mastectomy of initial RAASB (group A) did not undergo radical surgery including all pre-irradiated skin. Nevertheless, they showed local control (LC) for 37 and 67 months respectively, despite the fact that the latter patient had an R2 resection. In contrast to the review by Dogan et al. [8], this patient can be presented as a long-term survivor despite incomplete tumor resection. Thus, based on this limited and only single institutional experience, we would recommend adjuvant re-RT/HT, at least for cases where radical surgery is not possible or is refused.

Regarding local recurrences of RAASB as well as unresectable RAASB, limited data on treatment options and prognoses are available.

Standard treatment of locally recurrent RAASB is again surgical intervention with curative intent, whenever feasible. However, the rate of re-recurrences is high [5,15], often developing within a short period of time. Li et al. [11] report a median interval from first to second local recurrence of just 4 months. In contrast to the first RAASB resection, radical salvage surgery of recurrences did not significantly increase disease-specific survival [11].

In both first manifestations and recurrences of RAASB, prognosis is poor if R0 resection cannot be achieved [14]. Consequently, the same applies for unresectable RAASB. In a systematic review, long-term survivors were only found in cases of complete tumor resection [8].

De Jong et al. [21] and Linthorst et al. [22] reported their results of combined HT/re-RT for RAASB. Similar to our analysis, in both retrospective studies the percentage of patients with recurrent or unresectable disease was far above average compared to other single-institution reviews. The AMC protocol followed in Amsterdam consisted of a total dose of 32 Gy (8 × 4 Gy, 2×/week) combined with HT using a 434 MHz microwave technique given once per week within 1 h after RT [21]. Among 16 patients in AMC Amsterdam, 2 were unresectable, 12 were locally recurrent, and only 2 received combined adjuvant HT/re-RT after first resection. Complete response rate of macroscopic disease was 58%. Four of thirteen (31%) patients who were treated for macroscopic disease had good local control until either death or last follow-up (at 5, 7, 11, and 39 months) [21].

Of 23 patients at Erasmus MC Rotterdam, total RT doses ranged between 32 to 54 Gy (mean: 35 Gy) followed by HT either once a week for a total of four sessions or twice a week for a total of six sessions. HT was performed using 434 MHz microwave technique. Twelve patients presented with unresectable macroscopic disease. Complete response rate was 56%, and the 3-month, 1- and 3-year LC rates were 54%, 32%, and 22%, respectively. Eleven patients received adjuvant HT/re-RT after surgery, four with R0, six with R1, and one with R2 resection. Three-year LC rate was 46%. Late grade 4 RT toxicity was seen in two patients. One developed osteoradionecrosis of the chest wall requiring resection of the necrotic area, and the other required debridement for a chronic wound [22]. Molitoris et al. [38] published a case report on the treatment of a locally recurrent RAASB using a novel treatment intensification strategy including neoadjuvant HART with concurrent HT, followed by total mastectomy and flap reconstruction.

Our data are in complete agreement with the aforementioned data on beneficial effects of combined HT/re-RT in the treatment of recurrent and unresectable RAASB. Among three patients treated with adjuvant HT/re-RT after resection of recurrent RAASB (group B), one patient remained locally controlled during lifetime (R0 resection), and the other two patients who had an R1 resection, experienced a local recurrence. One of these two patients developed several re-recurrences, which could always be re-treated with the same schedule, leading to transient complete responses. The first local recurrence was in-field, and the others grew around the borders or out-of-field despite initial adequate RT- and HT-volumes (with margins of >5 cm up to 10 cm of all visible lesions). In contrast to our data on adjuvant re-RT/HT after surgery of first RAASB, re-recurrences following HT/re-RT after surgery of recurrent RAASB arise the question of higher adjuvant re-RT doses. However, intensification of re-RT dose is likely to be limited in such situations due to the presence of critical skin conditions. For example, patients No. 3 and 4 had skin transplantation applied on an already irradiated tissue (mesh graft, see Figure 3E and Figure 4B), therefore were at risk for critical toxicities of pre-irradiated or grafted tissue. Notably, none of the patients died from local progression. Even if no long-lasting LC was achieved, we have shown that this schedule can be used repeatedly, and is able to induce short-lasting CR, making this schedule a good palliative treatment option.

In cases of immediate macroscopic recurrence after surgery and refusal of another resection (group C), combined HT/re-RT can be utilized. The aforementioned data from the Netherlands, and the positive experience with patient No. 6 supports this.

The same applies for the palliative treatment of unresectable RAASB (group D). One heavily pretreated patient with extremely limited life expectancy did not respond to the treatment, however another patient with a large-sized RAASB achieved partial remission (PR) and good palliative effect. A patient with general good, condition despite an unresectable tumor, achieved a CR with local control (FU time 18 months). It is noteworthy that due to her excellent performance status, a standard fractionation with a semi-curative total dose was chosen.

The low acute and chronic toxicity rates upon a total re-RT dose of 20 Gy in five fractions (once weekly) may alleviate concerns of unacceptable toxicity of re-RT of RAASB. In fact, toxicity was low enough that even with repeat HT/re-RT with a time interval of only a few months, treatment could safely be delivered. In addition, the patient-friendly course of five weekly treatment sessions ensures a high compliance.

Note: Benign skin alterations, listed in detail in Table 2, also merit comment. In particular, the role of vascular degenerative changes, e.g., telangiectases (TA) and ruby points (cherry hemangioma) remain unclear in the development of RAASB. The incidence of TA has often been associated with RT-techniques, especially with the use of electrons ≥15 MeV. However, there are also dose-independent genetic risk factors as evidenced by the association of the XRCC1 (R399Q) DNA repair gene polymorphism with increased risk of telangiectases irrespective of the patient having received a boost [39]. Interestingly, our patients presented with TA outside of the former boost field in the remaining, previously irradiated skin post-mastectomy. This could support the hypothesis of additional gene factors. This observation is in agreement with Fodor et al. [40], who described post-RT related grade 2 TA in two out of eight patients. Brenn and Fletcher [41,42] associated TA with other post-radiation atypical vascular lesions (AVL) as a possible morphologic continuum in the development of cutaneous angiosarcoma. However, a clear correlation is missing, due to the small number of described cases. Ashack et al. [43] did not find any correlation of late RT-sequelae of skin and RAASB. Fair hair and skin color had been associated with the incidence of radiation induced basal cell carcinoma [44] but to our knowledge, it has never been as an important factor with regard to RAASB. Despite the occurrence of RAASB in our patients with fair hair, skin type I or II, the understanding of possible correlations is poor, due to the low number of patients. Banks et al. [45] reported skin fibrosis after previous RT in 19/56 patients. In our study 5 out of 10 patients presented with ruby points (cherry hemangiomas), and 2 with fibrosis of the remaining skin after mastectomy of RAASB. Another two patients had severe postoperative complications after breast conservation before the onset of RT (Table 1), but the fibrotic tissue was removed during the mastectomy. Based on these rare observations it is too early to give any recommendations to radiation oncologists for how to take care of AVLs at the onset of first RT of breast cancer. Nevertheless, we suggest noting cherry hemangiomas and other clinical observations in future prospective studies.

## 5. Conclusions and Outlook

Given the poor prognosis and limited treatment options in cases of recurrent or unresectable RAASB, our data and reports from the Netherlands suggest combined HT/re-RT as a beneficial palliative treatment option. In addition, combined HT/re-RT should be considered as an adjuvant treatment of RAASB when radical surgery, including all pre-irradiated skin areas, is not feasible. Low toxicity rates using thermography-controlled, contactless wIRA-hyperthermia immediately followed by hypofractionated re-RT (5 × 4 Gy treatments delivered once a week) allows for repeat re-irradiations in case of re-recurrences. The applied wIRA-hyperthermia method seems to be a good technical solution for heating RAASB patients, because it produces very large treatment fields. In addition, most of these tumors occur within the first 5 mm from the surface and can thus be heated adequately.

Prospective data on the use of combined HT/re-RT and corresponding multi-center studies are required.

## Figures and Tables

**Figure 1 cancers-13-03911-f001:**
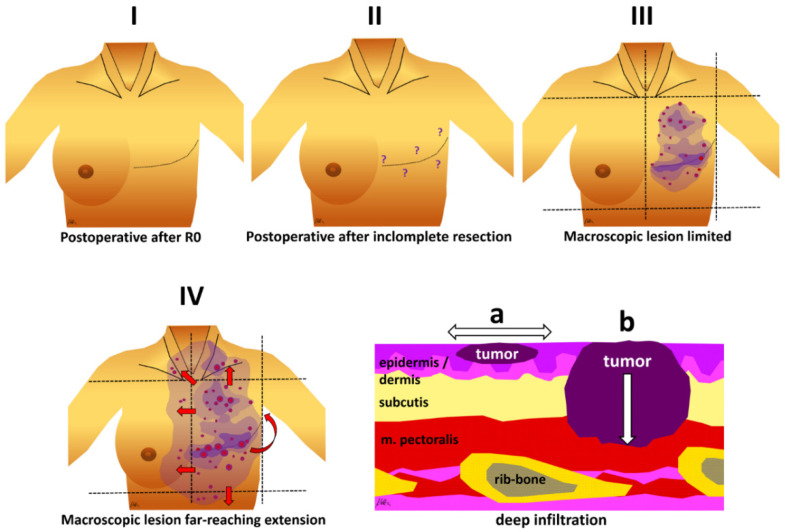
Different RAASB categories as used in this retrospective study. I = postoperative after R0, II = postoperative after incomplete resection (inappropriate margin, R1, or R2 respectively), III = macroscopic lesion limited to ipsilateral chest wall, not surpassing midsternal and midaxillary line, clavicle and lower costal margin, IV = macroscopic lesion with far-reaching extension, more than ipsilateral chest wall and supraclavicular fossa and/or contralateral chest wall or breast and/or abdominal wall and/or extension on the back. Lower right panel: Additional information on infiltration depth: a = infiltration limited to skin/subcutis, b = deep infiltration into muscle layers.

**Figure 2 cancers-13-03911-f002:**
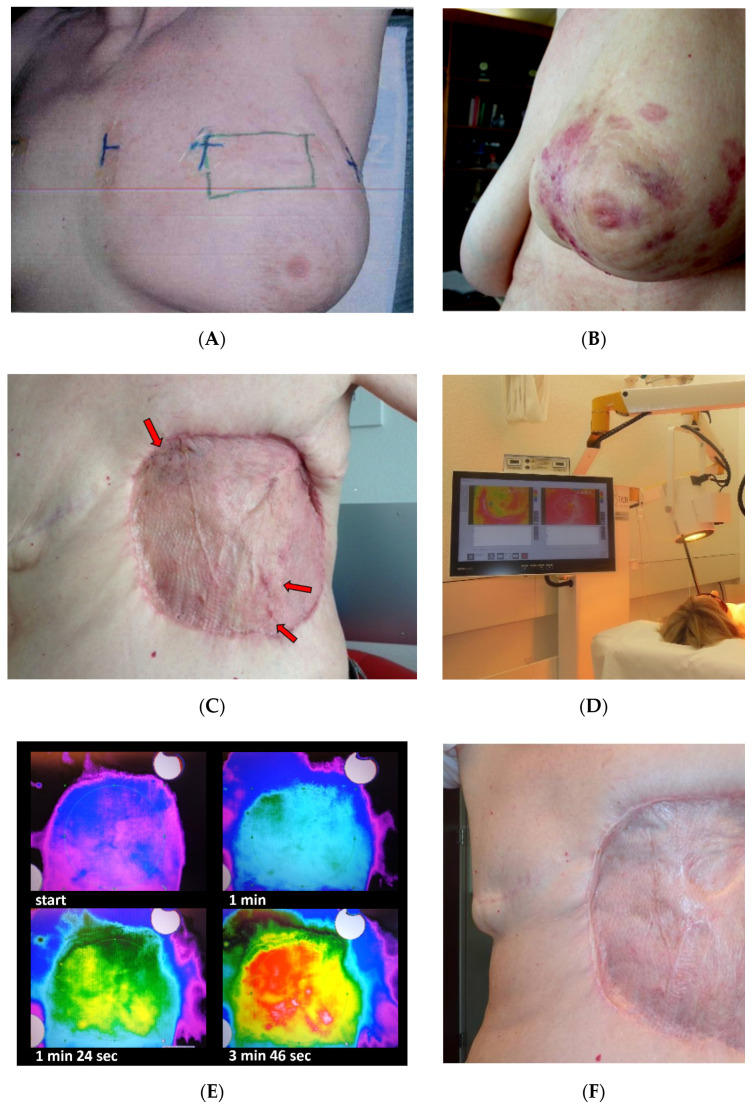
Documentation of patient No. 3. (**A**): Re-analysis of former radiotherapy 5 years prior to RAASB diagnosis: Field documentation of electron boost after tangential Cobalt-60 with 50 Gy (August 2006). (**B**): Initial RAASB, preoperative situation (May 2011). Note, that RAASB occurred outside former electron boost. (**C**)**:** Third local recurrence within the grafted skin (arrows, May 2014). (**D):** During treatment with hydrosun^®^TWH 1500. (**E)**: Corresponding thermography images during wIRA-hyperthermia: critical regions such as skin transplantation can be controlled visually and protected efficiently without causing burns. (**F**) Follow up 2.5 months after re-RT with 5 × 4 Gy + wIRA-hyperthermia 1×/week (September 2014).

**Figure 3 cancers-13-03911-f003:**
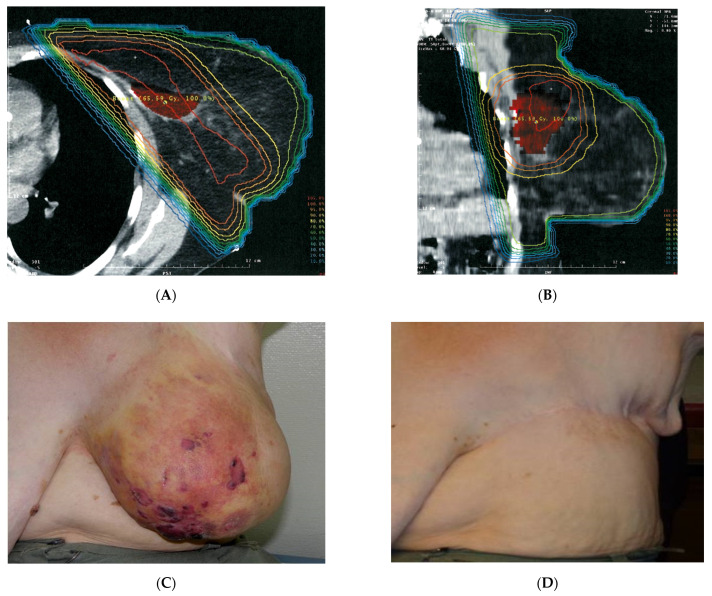
Documentation of patient No. 1. (**A**,**B**): Re-analysis of former RT 4 ½ years prior to RAASB diagnosis. 3D-planning of adjuvant breast irradiation (July 2007): Tangential irradiation of the breast with 50 Gy and boost to the former tumor bed with 16 Gy. Due to large diameter boost to the former, tumor bed was applied with parallel opposed photons field up to 16 Gy to minimize the risk of excessive skin fibrosis. (**C**): Initial RAASB: preoperative situation (August 2011). Note that the RAASB occurred all ower the breast and was not related to the former boost. (**D**): Follow up after incomplete resection (R2) and postoperative re-RT with 5 × 4 Gy + wIRA hyperthermia 1×/week (March 2014). Patient was lost to follow up after 67 months of observation. This exceptional case has been already published as a case report [31].

**Figure 4 cancers-13-03911-f004:**
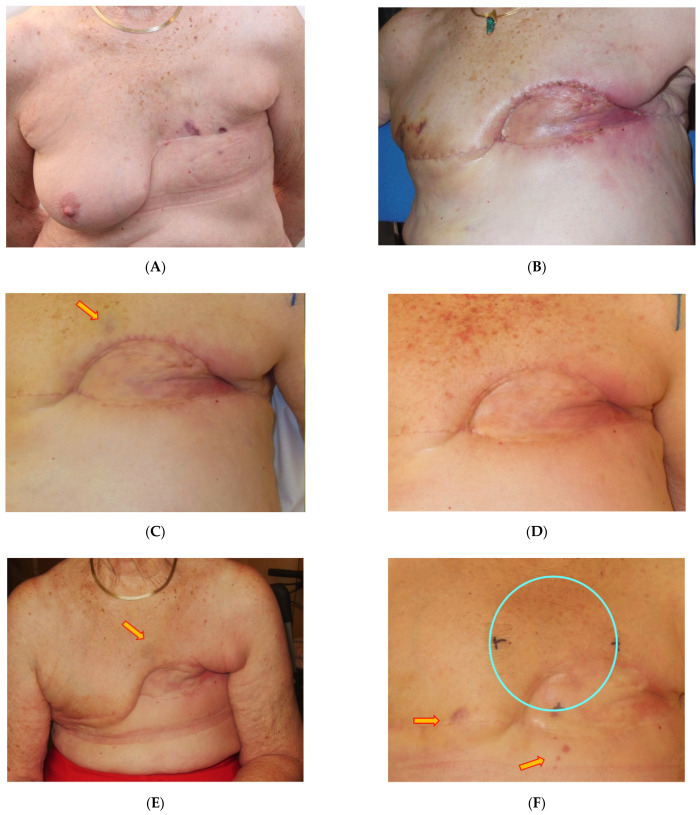
Successive photo documentation of patient No. 4: (**A**) 1st RAASB recurrence (March 3, 2017), after mastectomy 2015. (**B**) Postoperative situation including skin reconstruction with mesh graft, R1, and contralateral mastectomy (March 22, 2017) (**C**) Before start of HT/re-RT, detection of another new recurrence (April 2017, red arrow). (**D**) End of 1st series, complete remission (May 2017). (**E**) New recurrence inside (red arrow, June 2017): 2nd series of HT/re-RT, small field (June 2017). (**F**) End of 2nd series of HT/re-RT (August 2017): New PD (arrows) outside RT-field (blue circle). Patient cancelled further treatment due to personal circumstances. (**G**): Slow, but distinct progression, start of 3rd series (January 2018). (**H**) 5 months after 3rd series of HT/re-RT, complete remission (July 2018). (**I**) New PD inside/border/outside (red arrows). Start of 4th series (October 2018). (**J**) Middle of 4th series of HT/re-RT: distinct regression (November 2018). (**K**) 11 months after 4th series of HT/re-RT (October 2019). (**L**) FU, due to COVID-19 pandemic, patient could not present earlier: New PD outside field, slowly progressing (March 2021). Note occurrence of pronounced telangiectases, mainly limited to regions subjected to pressure by brassiere and breast prosthesis (additional mechanical factor).

**Figure 5 cancers-13-03911-f005:**
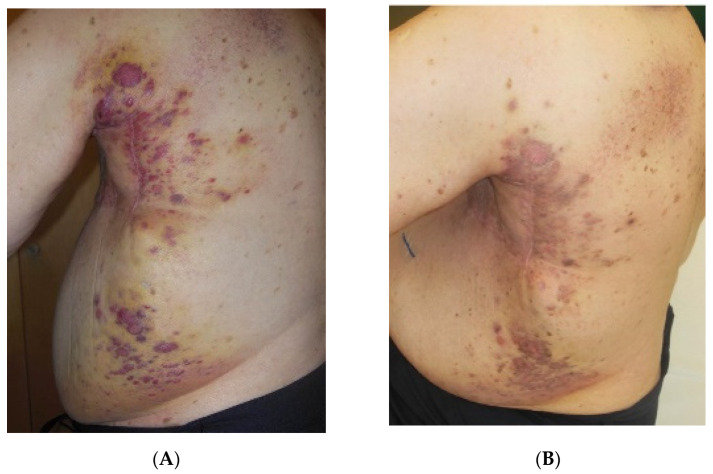
Successive documentation of Patient No. 9: (**A**) 1st consultation (February 2017). Recurrent RAASB with extension to the back growing along surgical scar (reconstruction of left chest wall). Size Class IV b. Palliative situation. (**B**) 3 weeks after wIRA-hyperthermia + 5 × 4 Gy (March 2017). PR in the treatment fields. Relief of pain.

**Table 1 cancers-13-03911-t001:** Former breast cancer treatment of patients with RAASB.

	Patient No. 1	Patient No. 2	Patient No. 3	Patient No. 4	Patient No. 5	Patient No. 6	Patient No. 7	Patient No. 8	Patient No. 9	Patient No. 10
**Age at Dx**	68	59	73	75	59	62	71	73	69	72
**BC-Stage**	T1N0M0	T1N0M0	T1N0M0	T1N0M0	T1N0M0	T2N0M0	T2N2aM0	T1N0M0	T2N0M0	T1N0M0
**Grade**	G2	G2	G1	G2	G2	G2	G2	G2	G2	G3
**BC-Tx**	BCS + RT + H	BCS + RT + H	BCS + RT + H	BCS + RT + H	BCS + RT + H	BCS + RT + H	BCS + RT + CT + H	BCS + RT + H	BCS + RT + H	BCS + RT + H
**RT Breast (Gy)**	25 × 2	28 × 1.8 *	25 × 2	25 × 2	21 × 1.8 ** + 6 × 2.2	25 × 2	28 × 1.8	25 × 2	28 × 1.8	25 × 2
**RT-Tech-nique**	Photons 6/15 MV	Photons 6 MV	^60^Co	Photons 6 MV	Photons 6 MV	Photons 6 MV	Photons 6 MV	Photons 6 MV	Photons 6 MV	Photons 6 MV
**Boost (Gy)**	8 × 2	Ø	5 × 2	5 × 2	5 × 2.5	5 × 2	5 × 2	5 × 2.5	8 × 2	8 × 2
**Type of Boost**	Photons 15 MV	Ø	e^−^ 9 MeV	e^−^ 9 MeV	Photons 6 MV	e^−^ 9 MeV	e^−^ 9MeV	e^−^ 9 MeV	e^−^ 12 MeV	e^−^ 9 MeV

Dx = diagnosis, BC = breast cancer, BC-Tx = breast cancer treatment, BCS = breast conserving surgery, RT = radiotherapy, CT = chemotherapy, H = hormonal therapy, * = postoperative RT delayed due to severe local infection of the reconstructed breast after surgery, ** = RT was interrupted for 5 weeks due to local infection of the breast, afterwards 2nd course of whole breast RT and additional boost, 6 resp. 15 MV = RT with linear accelerator, ^60^Co = RT with Cobalt-60 machine, Ø = no boost, e^−^ = electrons, MeV = energy of electrons.

**Table 2 cancers-13-03911-t002:** Clinical presentation of patients with RAASB.

	Patient No. 1	Patient No. 2	Patient No. 3	Patient No. 4	Patient No. 5	Patient No. 6	Patient No. 7	Patient No. 8	Patient No. 9	Patient No. 10
**Age at RAASB Dx**	73	63	80	80	71	68	78	80	74	86
**Latency BC-TTx to Dx of RAASB (y)**	5	4	7	4.5	12	6	7	6	7	14
**Initial RAASB Diameter (10 cm)**	>10	<10	<10	<10	<10	<10	<10	>10	>10	>10
**Grading**	3	3	2	3	3	2	3	3	3	3
**Skin Type**	II	I	II	II	II	II	I	I	I	I
**Hair**	brown	red	brown	fair	brown	brown	fair	fair	fair	fair
**Benign Skin Alterations**	Verrucae seniles	Lentigo solaris	Ruby points	Ruby points	Lentigo solaris		Ruby points	Ruby points	Ruby points	Lentigo solaris
**RT-related Skin Alteration**		Teleangi-ectases	Teleangi-ectases		Teleangi-ectases		Fibrosis lymph-edema		Fibrosis lymph-edema	

Clinical presentation and RT related skin alterations seen after surgery (mastectomy, further resections etc.) in the remaining previously irradiated skin. BC-Tx = breast cancer treatment, Dx = diagnosis.

**Table 3 cancers-13-03911-t003:** Grouping of patients with RAASB and treatment results.

	Patient No. 1	Patient No. 2	Patient No. 3	Patient No. 4	Patient No. 5	Patient No. 6	Patient No. 7	Patient No. 8	Patient No. 9	Patient No. 10
**Group**	**A**	**A**	**B**	**B**	**B**	**C**	**C**	**D**	**D**	**D**
**Surgery 1st RAASB**	TM	TM	TM	TM	Resect	TM	TM	TM	TM	TM
**R status 1st Surg**	R2	R0	R0	R0	R0	R0	R0	R0	R0	R0
**Latency to local recurrence**	-	-	8 months	10 months	52 months	4 weeks	3 weeks	1st: 4 months 2nd: 2 months 3rd: 3 months	3 months	14 months
**RAASB-recurrence: surgery**	-	-	Resect. (mesh graft)	Resect. (mesh graft)	Mast.	Ø	Ø	1st: resect 2nd: resect 3rd: inop.	Non-resectable	Non-resectable
**R status 2nd surgery**	-	-	R1	R1	R0	-	-	1st: R0 2nd: R1	-	-
**Chemotherapy**	Ø	Ø	2×	Ø	Ø	Ø	*	7×	Ø	Ø
**Tumor extension**	II	I	II	II	I	IIIa	IIIa	IIIb	IVb	IIIa
**HT/re-RT 1st course**	5 × 4 Gy 1×/w	5 × 4 Gy 1×/w	5 × 4 Gy 1×/w	5 × 4 Gy 1×/w	5 × 4 Gy 1×/w	5 × 4 Gy 1×/w	5 × 4 Gy 1×/w	5 × 4 Gy 1×/w + 2 × 4 Gy	5 × 4 Gy 1×/w	25 × 2 Gy + 5 × 2 Gy
**Response 1st course**	-	-	-	-	-	CR	CR	PD	PR	CR
**Repeated HT/re-RT**	-	-	1×	3×	-	-	-	-	-	-
**Response repeated HT/re-HT**	-	-	CR	3 × CR	-	-	-	-	-	-
**Time to local progression (mts)**	-	-	6	3	-	-	-	0 (PD)	1	-
**Survival (mts)**	67	37	20	45	10	16	1	1	7	18
**Status**	LFU	Living, NED	Dead	Living, SD	Living, NED	Living, NED	Living, NED	Dead	Dead	Living, NED

Group A: Directly treated with adjuvant HT/re-RT after 1st surgery of a newly diagnosed RAASB. Group B: Patients treated with adjuvant HT/re-RT after recurrence of RAASB initially treated with surgery alone. Group C: Patients treated with HT/re-RT after recurrences developing within 3–4 weeks after 1st resection despite R0 resection. Group D: Patients with non-resectable recurrences after one or several surgical treatments, treated directly with HT/re-RT. TM = total mastectomy, Mast. = modified mastectomy, Ø = not received, R status 1st Surg. = radicality after 1st surgical intervention, R status 2nd Surg. = radicality after 2nd surgical intervention, inop = inoperable recurrence, CT = chemotherapy, Tumor extension: classification at the presentation in radiation oncology department: I = postoperative after R0, II = postoperative after incomplete resection, III = macroscopic lesion limited to the ipsilateral chest wall, IV = macroscopic disease far-reaching extension, suffix a = non deep infiltration, suffix b = deep infiltration, re-RT = re-irradiation, HT = hyperthermia, Tx = treatment, LFU = Lost to follow up,* = patient presents distant metastasis of breast cancer under systemic treatment, together with palliative radiation of different manifestations, therefore no further extensive surgery.

**Table 4 cancers-13-03911-t004:** Acute and chronic side effects after HT-re-RT.

	Patient No. 1	Patient No. 2	Patient No. 3	Patient No. 4	Patient No. 5	Patient No. 6	Patient No. 7	Patient No. 8	Patient No. 9	Patient No. 10
**Acute side effects**										
Erythema	G1	G1	G1	-	G1	G1	G1	-	-	G1
Hyper-pigmentation	-	G1	G1	-	G1	G1	-	G1	G1	-
**Chronic side effects**										
Induration	-	-	-	-	-	-	-	-	-	-
Hyper-pigmentation	-	-	-	G1	-	-	-	-	-	G1
New telangiectases	-	-	-	G1 *	-	-	-	-	-	-

Acute and chronic side effects are indicated according to CTCAE grading. * = new telangiectases after 4 series of re-RT/HT with 5 × 4 Gy 1x/week, mainly in mechanically charged skin parts. No thermal related skin damage, e.g., blisters, has been noted. Grade 1 (G1) toxicity was handled with usual topical skin treatment.

## Data Availability

Data supporting reported results are in the archives of the hospital and not available for any outside institution due to legal regulations. In addition, anonymous data of the patients are saved in cantonal or federal cancer registries, but again not available publicly.

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
