# Peer review of "Radiation-Associated Angiosarcoma of the Breast and Chest Wall Treated with Thermography-Controlled, Contactless wIRA-Hyperthermia and Hypofractionated Re-Irradiation"

_cancers, 2021, doi:10.3390/cancers13153911_

Round 1
Reviewer 1 Report
The authors report a retrospective cohort of patients with RAASB after adjuvant radiation treatment of the breast. RAASB is a pretty rare disease, and thus investigations of treatment options are required. Especially the presented hypofractionated re-irradiation schedule with an application of wIRA-hyperthermia to reduce the radiation dose seems interesting. Authors report the treatment results of 10 patients treated between 2011-2015.
The authors drafted a well written, concise, detailed, and clearly structured article with extensive illustrations.
Limitations:
- ten retrospective analysed patient cases (low case number)
- only descriptive reporting of outcome and toxicity possible, no statistical analysis
-
no prospective protocol, and thus heterogeneous cohort with different technique, dose and treatment schedules (adjuvant after RAASB surgery, treatment after recurrence surgery of RAASB, unresectable recurrence…)
-
The abstract seems to be incomplete, with no information on results and no conclusion.
Author Response
Reviewer 1:
Comments and suggestions for the authors:
The authors report a retrospective cohort of patients with RAASB after adjuvant radiation treatment of the breast. RAASB is a pretty rare disease, and thus investigations of treatment options are required. Especially the presented hypofractionated re-irradiation schedule with an application of wIRA-hyperthermia to reduce the radiation dose seems interesting. Authors report the treatment results of 10 patients treated between 2011 and 2015.
The authors drafted a well written, concise, detailed, and clearly structured article with extensive illustrations.
Reply:
Thank you for your supportive remarks
Limitations:
- Remark:
Ten retrospective analysed patient cases (low case number)
- Answer:
Indeed, 10 cases would be a low number for a common tumor. But RAASB is a rare tumor entity with a prevalence of approx. 1/1000 patients treated with radiotherapy for breast cancer. Further, our study describes a new innovative approach regarding RT fractionation and application method of hyperthermia showing remarkable results. This is a retrospective study but treatment reports, side effects, outcomes and further interventions and its results have been reported very precisely in an "extended case report" manner. This allows the readers to learn more about the possible behavior of this rare disease. We, therefore, think that even this low case numbers justifies publication in a highly ranked journal like "Cancers".
- Remark:
Only descriptive reporting of outcome and toxicity possible, no statistical analysis
- Answer:
We completely agree that statistics do not make sense when presenting 10 cases
- Remark:
No prospective protocol, and thus heterogeneous cohort with different technique, dose and treatment schedules (adjuvant after RAASB surgery, treatment after recurrence surgery of RAASB, unresectable recurrence…)
- Answer:
A prospective protocol in such a rare disease with an additional heterogeneous clinical presentation is very difficult. We think that the first step would be a retrospective analysis of patients treated in an individual approach as we show in the present manuscript. Its results shed more light on the treatment of this disease and lead to the classification of different treatment situations, as indicated in Tab. 3: A: adjuvant re-RT & HT after RAASB surgery, B: adjuvant re-RT & HT after surgery of recurrent disease, C: re-RT & HT in recurrent RAASB after surgery within short time, and D: re-RT & HT in non-operable (recurrent) RAASB. These results and this classification provide the platform for initiating a prospective, multicenter single arm protocol to validate this treatment schedule. With one exception, all patients have been treated with the same treatment schedule (5 x 4 Gy 1x/week, immediately after superficial HT)
- Remark
The abstract seems to be incomplete, with no information on results and no conclusion
- Answer:
Thank you for pointing this out. We have restructured the abstract and added treatment results and conclusions more precisely as follows:
BACKGROUND: Radiation-associated angiosarcoma of the breast (RAASB) is a rare, challenging disease, with surgery being the accepted basic therapeutic approach. In contrast, the role of adjuvant and systemic therapies is controversially discussed. Local recurrence rates reported in the literature are mostly heterogeneous and are highly dependent on the extent of surgery. In cases of locally recurrent or unresectable RAASB, prognosis is very poor.
METHODS: We retrospectively report on 10 consecutive RAASB patients, most of them presenting with locally recurrent or unresectable RAASB, which were treated with thermography-controlled water-filtered infrared-A (wIRA) superficial hyperthermia immediately followed by re-irradiation. Patients with RAASB were graded based on their tumor extent before onset of RT.
RESULTS: We recorded a local control (LC) rate dependent on tumor extent ranging from a high LC rate of 100% (2/2 patients) in the adjuvant setting with a R0 or R2 resection to a limited LC rate of 33% (1/3 patients) in patients with inoperable, macroscopic tumor lesions.
CONCLUSION: Combined hyperthermia (HT) and re-irradiation (re-RT) should be considered as an option for (a) adjuvant treatment of RAASB, especially in cases with positive resection margins and after surgery of local recurrence (LR), and (b) for definitive treatment of unresectable RAASB.
Thank you for your valuable comments and suggestions
Reviewer 2 Report
This is a well written case report on a limited number of patients, treated with heterogeneous techniques. But, seen the fact that it concerns a rare disease with poor outcome, information on innovative approaches is of interest to the radiotherapy (surgery) community.
I have no major remarks, but a lot of the the patient information (and Table/figure 1) belong in my opinion to the result section
Author Response
Reviewer 2:
Comments and suggestions: This is a well written case report on a limited number of patients, treated with heterogeneous techniques. But, seen the fact that it concerns a rare disease with poor outcome, information on innovative approaches is of interest to the radiotherapy (surgery) community.
Reply:
Thank you for your supportive remarks.
Remark:
I have no major remarks, but a lot of the patient information (and Table/Figure 1) belong in my opinion to the result section
- Answer:
Concerning Table. 1:
Thank you for to precise this issue. All informations (age at diagnosis, TNM-stage, grading, surgery, hormonal treatments and radiotherapy, which is even described in detail) are related to the former primary treatment of breast conserving therapy, and are not related to the treatment of RAASB. Fig. 1 is related to the attempt to describe RAASB extensions before start of re-RT and HT. We therefore would like to keep Table 1/Figure 1 within the patient information in the "2. Patients and methods" section of the manuscript.
But we would like to make the title of this table clearer by changing it from Tab. 1: old text: Treatment of patients presenting with RAASB, new text: Former breast cancer treatment of patients with RAASB
Thank you for your valuable comments and suggestions
Reviewer 3 Report
The authors present a retrospective study about the experience in treating 10 patients with a radiation induced angiosarcoma with re-irradiation and hyperthermia using thermography controlled infrared-A hyperthermia. The study is well-written and gives a complete and concise overview of the patient, tumors, treatment and outcomes.
I only have a few minor comments:
Abstract
please provide a general conclusion about the clinical outcome of the 10 patients treated with re-irradiation and hyperthermia. This should match the recommendentations for treatment made in the abstract.
Introductio
please provide additional on the technique of infrared-A superficial hyperthemia; to what depth can the hyperthermia be applied? Potential side effects, burns? And how to handle these?
Patients and methods:
Table 1 seems a bit redundant as all patients received previous irradiation.
Table 3: it is visually not clear how the groups A-D are divided in the table, suggestion to make grouped colums
lines 161 - 16: please elaborate how the hyperthermia field relates to the radiation field in terms of depths; are some parts of the tumor not reached by the hyperthermia?
Results: Fig 2a an 2b seem a bit redundant as no conclusion can be drawn about the relation of the location of the RAASB and the previous radiation field
Discussion:
lines 304-305 and 320-323: late grade IV toxicity was limited in the Rotterdam study; why do the authors think it will be more critical in their patients?
Author Response
Reviewer 3:
Comments and suggestions:
The authors present a retrospective study about the experience in treating 10 patients with a radiation induced angiosarcoma with re-irradiation and hyperthermia using thermography- controlled infrared-A hyperthermia. The study is well-written and gives a complete and concise overview of the patient, tumors, treatment and outcomes.
Thank you for your supportive remarks.
I have only a few minor comments:
Abstract:
Please provide a general conclusion about the clinical outcome of the 10 patients treated with re-irradiation and hyperthermia. This should match the recommendations for treatment made in the abstract.
Reply:
Thank you for pointing out that in the abstract general conclusions about the clinical outcome should be stated. We have restructured the abstract and added treatment results and conclusions more precisely as follows:
BACKGROUND: Radiation-associated angiosarcoma of the breast (RAASB) is a rare, challenging disease, with surgery being the accepted basic therapeutic approach. In contrast, the role of adjuvant and systemic therapies is controversially discussed. Local recurrence rates reported in the literature are mostly heterogeneous and are highly dependent on the extent of surgery. In cases of locally recurrent or unresectable RAASB, prognosis is very poor.
METHODS: We retrospectively report on 10 consecutive RAASB patients, most of them presenting with locally recurrent or unresectable RAASB, which were treated with thermography-controlled water-filtered infrared-A (wIRA) superficial hyperthermia immediately followed by re-irradiation. Patients with RAASB were graded based on their tumor extent before onset of RT.
RESULTS: We recorded a local control (LC) rate dependent on tumor extent ranging from a high LC rate of 100% (2/2 patients) in the adjuvant setting with a R0 or R2 resection to a limited LC rate of 33% (1/3 patients) in patients with inoperable, macroscopic tumor lesions.
CONCLUSION: Combined hyperthermia (HT) and re-irradiation (re-RT) should be considered as an option for (a) adjuvant treatment of RAASB, especially in cases with positive resection margins and after surgery of local recurrence (LR), and (b) for definitive treatment of unresectable RAASB.
Introduction:
Please provide additional on the technique of infrared-A superficial hyperthermia; to what depth can the hyperthermia be applied? Potential side effects, burns? And how to handle these?
Reply:
We have described the technique of infrared-A-superficial hyperthermia in section "2.2. Treatment" and not in the "Introduction" section. For further information, the technique has been described in detail in Vaupel P. et al. in: "Biophysical and photobiological basis of water-filtered infrared-A-hyperthermia of superficial tumors", Int. J. Hyperthermia, 2018, 35(1), 26-36.
But we see your point that some technical information would be helpful being provided directly in the introduction section. Therefore, we applied your suggestions by adding the following information:
Line 71: This hyperthermia technique guarantees large treatment fields, tissue temperatures >40°C up to 15 mm, and only low toxicity (mostly Grade 1) due to permanent temperature measurements all over the body surface avoiding any burns [30].
Because there are no burns seen in our patients (see Table 4), handling of G1 side effects was not described in detail due to the length of this drafted article. But we applicate your suggestion and add in the text of Table 4 the following information:
Text of Table 4:
Acute and chronic side effects are indicated according to CTCAE grading. * = new teleangiectases after 4 series of re-RT/HT with 5 x 4 Gy 1x/week mainly in mechanically charged skin parts. No thermal related skin damage, e.g., blisters, has been noted. G1 toxicity was handled with usual topic skin treatment.
Patients and methods:
Table 1 seems a bit redundant as all patients received previous irradiation.
Reply:
Thank you for this point. We agree, that radiation associated tumors will always have in their history radiation exposure. But we think, that detailed information of former radiation treatment in RAASB is important specifically analyzing the radiation technique used. Furthermore, the question of boost or no boost and the application of electrons is still an open question in the occurrence of RAASB, as mentioned by some authors (e.g., 38, 39). However, in order to reduce confusion, we change the title of Table 1 as follows: old text: Treatment of patients presenting with RAASB, new text: Former breast cancer treatment of patients with RAASB.
Therefore, this table serves as a base of patient characteristics.
Table 3: it is visually not clear how the groups A-D are divided in the table, suggestion to make grouped cohorts.
Reply:
Thank you very much for raising this important point. We completely agree with you. In our uploaded draft we had vertical columns to separate the different groups, but probably due to formatting issues, the preprint draft appeared differently. We had already informed the editors concerning this issue. Supported by your remark we would discuss the format issue again with the editors. You can see the original Table 3 below:
Table 3
|
Pat N° 1 |
Pat N° 2 |
Pat N° 3 |
Pat N° 4 |
Pat N°5 |
Pat N° 6 |
Pat N° 7 |
Pat N° 8 |
Pat N° 9 |
Pat N°10 |
|
|
Group |
A |
B |
C |
D |
||||||
|
Surgery 1st RAASB |
TM |
TM |
TM |
TM |
Resect |
TM |
TM |
TM |
TM |
TM |
|
R status 1st Surg |
R2 |
R0 |
R0 |
R0 |
R0 |
R0 |
R0 |
R0 |
R0 |
R0 |
|
Latency to local recurrence |
- |
- |
8 mts |
10 mts |
52 mts |
4 weeks |
3 weeks |
1st: 4 mts 2nd: 2 mts 3rd : 3 mts |
3 mts |
14 mts |
|
RAASB-recurrence: surgery |
- |
- |
Resect. (mesh graft) |
Resect. (mesh graft) |
Mast. |
Ø |
Ø |
1st: resect 2nd: resect 3rd: inop. |
Non-resectable |
Non-resectable |
|
R status 2nd surgery |
- |
- |
R1 |
R1 |
R0 |
- |
- |
1st: R0 2nd: R1 |
- |
- |
|
Chemotherapy |
Ø |
Ø |
2x |
Ø |
Ø |
Ø |
* |
7x |
Ø |
Ø |
|
Tumor extension |
II |
I |
II |
II |
I |
IIIa |
IIIa |
IIIb |
IVb |
IIIa |
|
HT/re-RT 1st course |
5 x 4 Gy 1x/w |
5 x 4 Gy 1x/w |
5 x 4 Gy 1x/w |
5 x 4 Gy 1x/w |
5 x 4 Gy 1x/w |
5 x 4 Gy 1x/w |
5 x 4 Gy 1x/w |
5 x 4 Gy 1x/w + 2 x 4 Gy |
5 x 4 Gy 1x/w |
25x2 Gy+ 5x2 Gy |
|
Response 1 st course |
- |
- |
- |
- |
- |
CR |
CR |
PD |
PR |
CR |
|
Repeated HT/re-RT |
- |
- |
1x |
3 x |
- |
- |
- |
- |
- |
- |
|
Response repeated HT/re-HT |
- |
- |
CR |
3 x CR |
- |
- |
- |
- |
- |
- |
|
Time to local progression (mts) |
- |
- |
6 |
3 |
- |
- |
- |
0 (PD) |
1 |
- |
|
Survival (mts) |
67 |
37 |
20 |
45 |
10 |
16 |
1 |
1 |
7 |
18 |
|
Status |
LFU |
Living, NED |
Dead |
Living, SD |
Living, NED |
Living, NED |
Living, NED |
Dead |
Dead |
Living, NED |
Lines 161 – 166: please elaborate how the hyperthermia fields relates to the radiation field in terms of depths: are some parts of the tumor not reached by the hyperthermia?
Reply:
Thank you for this very important issue. You point out the key question that every clinical radiation oncologist has to assure the right depth of planned radiation field and related hyperthermia fields. Your suggestion gives us a chance to provide more details about our planning of hyperthermia and re-irradiation. Base was the CT scan (planning CT) and former radiological investigations such as MRI’s, CT and PET-CT’s and clinical descriptions as well. Knowing the difficulties to define properly margins around RAASB lesions with the very often observed occult metastasis (like "flee spread") in neighboring normal looking skin parts we prescribed at least 5 – 10 cm margins. That means that RT-fields, mostly using electrons, were quite large and mainly limited by the machines’ capability. We chose electrons because RAASB is mainly a skin disease occurring in cutaneous/subcutaneous layers. This led us to the proposal to define properly tumor extension at beginning of re-irradiation: once in horizontal direction (see Fig 1 categories I – IV) and on the other side deep infiltration (see Fig 1 category additional information a or b). Using this classification in our patients, it is clearly visible, that only patient 8 and 9 presented deep infiltration (see Table 3), in both not more than 2.0, resp. 2.5 cm max, because the recurrent RAASB were located after previous mastectomy in the chest wall. One important technical advantage of infrared water filtered hyperthermia is to produce very large treatment fields, mostly even larger than radiation fields (see also the descriptions in Vaupel P. et al., Int. J. Hyperthermia, 2018, 35(1), 26-36.). This technique achieves > 40°C in tissue depth up to approximatively 1.5 cm, > 39°C in up to approximatively 2.0 cm. Contact free temperature measuring by thermography solves the difficulties to control the whole surface. To conclude we think that this technique is ideal for heating RAASB patients, because most of these tumors occur within the first 5 mm. In addition, major concern is not the depth of hyperthermia or radiation, but the field size. Therefore, we changed the description of the RT and HT-treatments as follows (lines 154 – 169)
Treatment
Treatment planning was based on CT scan, former radiological investigations, e.g., MRI’s and PET-CT and clinical descriptions to define properly lateral tumor spread and deep infiltration. RT- and HT-volumes were prescribed with margins of 5 to 10 cm around all visible lesions whenever possible. That means that RT-fields, mostly using electrons, were quite large and mainly limited by the machines’ capability. Electrons were chosen because RAASB is considered a skin disease occurring in cutaneous/subcutaneous layers. The treatment protocol consisted of weekly water filtered infrared-A (wIRA) superficial HT for 45 – 60 min (hydrosun®TWH1500, Hydrosun Medizintechnik, Müllheim, Germany, see Fig. 3D), immediately followed by re-RT. The contactless energy deposition allows for continuous thermography as well as visual control of the entire treatment field, which is of crucial importance for critical skin conditions, e.g., skin transplants and mesh grafts (Figs. 3C, 3F). In addition, this technique produces very large treatment fields [30], mostly even larger than radiation fields in order to cover completely the defined treatment volume. The computer-based, closed feedback system of this device was set at a maximum skin surface temperature of 43.0°C (Fig. 3E). This results in tissue temperatures > 40°C at a tissue depth up to approximately 15 mm, and > 39°C up to a depth of 25 - 30 mm [30]. Patients N° 1-9 were treated with 4 Gy once per week for a total dose of 20 Gy, as described in detail later [28,29,30]. Eight patients were irradiated with 6 – 9 MeV electrons and a preheated bolus. Patient N° 8 had a combined photon-electron plan to cover deep infiltration. Patient N° 10 received a conventionally fractionated re-RT with 50 Gy (25 fractions with 2 Gy/fraction with photons) with an electron boost up to 60 Gy which was combined with 6 weekly hyperthermia sessions, applied immediately before RT.
Again, thank you to bring us to a much better description of the applied method. In addition, we add the aforementioned technical advantages of wIRA-hyperthermia in the conclusion as follows:
Line 378: The applied wIRA-hyperthermia method seems to be a good technical solution for heating RAASB patients, because it produces very large treatment fields. In addition, most of these tumors occur within the first 5 mm from the surface and can thus be heated adequately.
Results: Fig 2a and 2b seem a bit redundant as no conclusion can be drawn about the relation of the location of the RAASB and the previous radiation field.
Reply:
We agree, that at a first glance the value of Figs. 2a and 2b do not seem to be very informative. You argue, that no correlation of former RT field and location of RAASB can be drawn. In our opinion that’s the key point. RAASB in our patients didn’t not occur more frequently in previous boost regions, either in former electron- or photons fields. Just to note, that skin fibrosis as late RT sequelae is associated in case of larger breasts, higher Dose max’s (Dmax) of radiation doses and fractionation. Patient 1 had a rather large diameter of her breast forcing the former radiation oncologist to apply a parallel opposed photon boost. Because it is an "extended case report series" we propose to keep this illustration, but comment it more precisely:
Fig. 2. Documentation of patient N° 1. (A, B): Re-analysis of former RT 4.5 years prior to RAASB diagnosis. 3-D-planning of adjuvant breast irradiation (July 2007): Tangential irradiation of the breast with 50 Gy. Due to large diameter boost to the former tumor bed was applied with parallel opposed photons field up to 16 Gy to minimize the risk of excessive skin fibrosis. (C): Initial RAASB: preoperative situation (August 15, 2011). Note that the RAASB occurred all ower the breast and was not related to the former boost region. (D): Follow up after incomplete resection (R2) and postoperative re-RT with 5 x 4 Gy + wIRA hyperthermia 1x/week (March 14, 2014). Patient was lost to follow up after 67 months of observation. This exceptional case has been already published as a case report [45].
Discussion:
Lines 304 – 305 and 320-323: late grade IV toxicity was limited in the Rotterdam study, why do the authors think it will be more critical in their patients?
Reply:
It was not at all our aim to give the impression, that results in Rotterdam were worse. The communicated side effects just demonstrate how critical re-treatment specifically in recurrent RAASB can be. We only wanted to describe the two G IV events for uninformed readers. We don’t think, that such an event would be more critical in our patients. Such a severe toxicity would be critical for every patient, but has to be counterbalanced against the possible therapeutic gain. Considering postoperative situation with skin transplants on previously irradiated tissue ground every local retreatment has the potential of severe side effects. May be, that the applied treatment schedule in our study does have the advantage of better tolerance due to the low total dose. Up to now, also in analogous situations, e.g., locally recurrent breast cancer we never observed up to now any severe toxicity, but we are aware, that this can never completely be excluded and patients are always informed about such possible adverse effects as well.
Thank you for your valuable comments and suggestions.
Reviewer 4 Report
In this manuscript authors describe 10 patients affected by radiation-induced angiosarcoma of the breast/thoracic wall treated with re-irradiation (1. RT course was performed after surgery at the first breast cancer diagnosis) combined to hyperthermia. Even though radiation-induced angiosarcomas are very rare, there is a clinical interest for these kinds of cases, since their prognosis is poor and treatment options very limited. The manuscript is clear and well written, even if too long, in my opinion. Nevertheless, since with 10 cases any statistical analysis is nonsense, I agree with the authors that a description of treatments and outcomes practically for every patient is the best way to present the data. For these reasons, I have no additional requests of changes and support therefore the publication of the present manuscript in Cancers in the Hyperthermia dedicated section.
Author Response
Reviewer 4:
Comments and suggestions:
In this manuscript authors describe 10 patients affected by radiation-induced angiosarcoma of the breast/thoracic wall treated with re-irradiation (1. RT course was performed after surgery at the first breast cancer diagnosis) combined to hyperthermia. Even though radiation-induced angiosarcomas are very rare, there is a clinical interest for these kinds of cases, since their prognosis is poor and treatment options very limited. The manuscript is clear and well written, even if too long, in my opinion. Nevertheless, since with 10 cases any statistical analysis is nonsense, I agree with the authors that description of treatments and outcomes practically for every patient is the best way to present the data. For these reasons, I have no additional requests of changes and support therefore the publication of the present manuscript in Cancers in the hyperthermia section.
Reply:
Thank you for your supportive remarks.
Round 2
Reviewer 1 Report
Thank you for the revisions.